# Real Coded Mixed Integer Genetic Algorithm for Geometry Optimization of Flight Simulator Mechanism Based on Rotary Stewart Platform †

**Miloš D. Petrašinović *** , **Aleksandar M. Grbović** , **Danilo M. Petrašinović** , **Mihailo G. Petrović** and **Nikola G. Raičević**

Faculty of Mechanical Engineering, University of Belgrade, Kraljice Marije 16, 11000 Belgrade, Serbia; agrbovic@mas.bg.ac.rs (A.M.G.); dpetrasinovic@mas.bg.ac.rs (D.M.P.); petrovicmiha@gmail.com (M.G.P.); nraicevic@mas.bg.ac.rs (N.G.R.)
* Correspondence: mpetrasinovic@mas.bg.ac.rs; Tel.: +381-60-0810-994
† This paper is an extended version of our paper published in Computational and Experimental Approaches in Materials Science and Engineering, Zlatibor, Serbia, 2–5 July 2019.

**Featured Application: Low-cost flight simulators with electric rotary actuators and optimized geometry for flight simulation.**

**Abstract:** Designing the motion platform for the flight simulator is closely coupled with the particular aircraft's flight envelope. While in training, the pilot on the motion platform has to experience the same feeling as in the aircraft. That means that flight simulators need to simulate all flight cases and forces acting upon the pilot during flight. Among many existing mechanisms, parallel mechanisms based on the Stewart platform are suitable because they have six degrees of freedom. In this paper, a real coded mixed integer genetic algorithm (RCMIGA) is applied for geometry optimization of the Stewart platform with rotary actuators (6-RUS) to design a mechanism with appropriate physical limitations of workspace and motion performances. The chosen algorithm proved that it can find the best global solution with all imposed constraints. At the same time, the obtained geometry can be manufactured because integer solutions can be mapped to available discrete values. Geometry is defined with a minimum number of parameters that fully define the mechanism with all constraints. These geometric parameters are then optimized to obtain custom-tailored geometry for aircraft flight simulation.

**Keywords:** real coded mixed integer genetic algorithm; flight simulator; rotary Stewart platform; motion platform; geometry optimization



## 1. Introduction

From the very beginning of aviation, in addition to the construction of the aircraft, the most demanding task was to master the aircraft piloting. The rapid development of civil aviation soon required the possibility of instrumental flying. Pilots had to be able to control the aircraft in conditions when visibility from the cockpit was so reduced that the flight was no longer safe. In 1929, to reduce the risk and cost of training, the American constructor Edwin A. Link constructed the Link Trainer flight simulator. The cockpit was mounted on a carrier which enabled change of position and orientation, depending on the given command, in the same way as a real aircraft. It is also crucial to have the same instruments which react accordingly.

Today, flight simulators play an irreplaceable role in pilot training and certification. Their full potential was discovered when there was not enough time to train a large number of pilots on real planes. With these machines, pilots could learn how to fly under conditions in which flight by visual reference is not safe anymore. In such situations, the pilot needs to

know how to control the aircraft, relying only on information collected from the instruments and its own vestibular system.

Over time, flight simulators have become complex mechatronic systems, but their main purpose, to enable pilots to feel what they see with their eyes, remains the same. In order to achieve realistic simulation in all flight cases and all forces acting upon the pilot during the flight, a moving part of the simulator needs to have six degrees of freedom. The chosen mechanism with six degrees of freedom has to have a sufficiently large workspace to simulate aircraft motion, which is why the determination of motion boundaries is very important.

A free rigid body has six degrees of freedom, it can perform six independent motions, three translations and three rotations. A kinematic chain is a series of bodies or segments connected together by joints. A kinematic chain can be simple or complex, and both can be divided into open and closed. One segment of a simple kinematic chain cannot be connected to more than two other segments, while one segment of a complex kinematic chain can be connected to more than two other segments. Open kinematic chains have segments that are connected to only one other segment, while all closed chain segments are connected to at least two other segments, as shown in Figure 1.

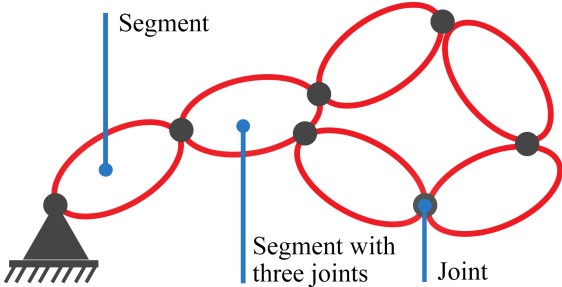

**Figure 1.** Closed kinematic chain.

Among many existing designs, parallel mechanisms (mechanisms with closed chains) based on the Stewart platform, defined in [1], are most frequently used for flight simulators. From this first work until today, this mechanism is a popular research topic in robotics with many research papers published about inverse and forward kinematics problems [2,3], dynamics [2,4–7], singularities and workspace estimation [2,8–11]. In a paper [12] written in 1998, the authors presented a state-of-the-art review of the literature on the Stewart platform and mentioned papers are still the basis of research. A similar overview paper was written in 2016 [13].

Even after so many years (almost six decades) of research, there is still the opportunity to find a new or better way of practical use of this mechanism. Besides flight simulators, this mechanism is currently used, for instance, for machining tools [14], as part of medical instruments for precision surgeries [15], for high-precision vibration damping for spacecraft payload [16], within marine satellite tracking antenna stabilization system [17], for non-destructive inspection tool [18], for assembly of flaps [19], for testing of micro-electro-mechanical system dynamic inclinometer [20], for energy-efficient machining bed [21], gait rehabilitation [22,23], stabilization system for optoelectronic devices [24], docking mechanism [25] and even for helicopter floating helideck [26]; additional applications are shown in [27].

There are many different types of geometrical interpretations of the Stewart platform, but the main objective of this paper is geometry optimization of the Stewart platform with rotary actuators, as shown in Figure 2. It has six pairs of upper and lower levers (together called legs), and the movement of the upper platform relative to the base is achieved by rotating the lower lever around an axis going through one end of a lever. This is the revolute joint between the base and a lower lever controlled by an electric motor. The upper and lower levers are connected with universal or spherical joints, which connect the upper lever and platform. This is why this type is also called 6-RUS, in accordance with the types of joints.

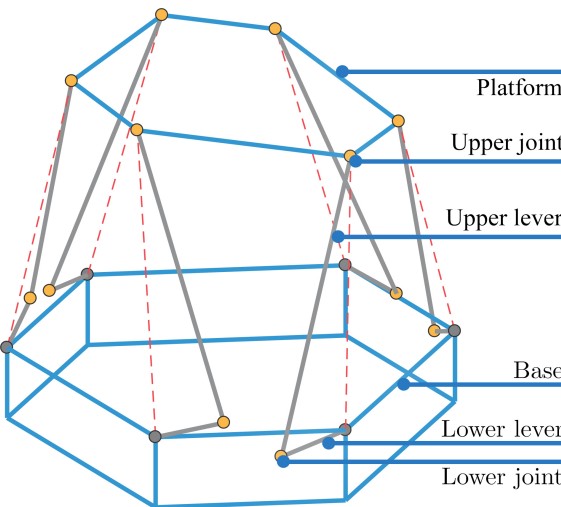

**Figure 2.** Stewart platform with rotary actuators.

The contribution of this paper is significant as it provides solutions to both inverse and forward kinematics problems for the rotary Stewart platform, besides the geometrical definition of the mechanism and algorithm for its optimization.

To control the position and orientation of the platform, the part of the simulator that connects the base and platform (leg) needs to have an actuator. Some of the other geometrical types use six levers with variable lengths (linear or prismatic actuators) or six pairs of two levers, of which one lever is fixed but has a variable length. The most critical design aspects of the mechanism besides the types of actuators are spatial configuration (locations of connections) and the type of connections (joints).

A design solution based on rotary actuators, in this case, electric motors with proper gearboxes, is adopted due to its simplicity in terms of simulation, production, and maintenance. If the servo motor with servo drive is not cost-effective, an induction motor or asynchronous motor can be controlled using a variable-frequency drive (VFD) and both products are standardized and widely available. The current angle of the shaft can be measured with an encoder or potentiometer. The mechanism with this type of actuator is the simplest one to scale down and test on a smaller model. Even before investing in the first prototype, kinematics and control algorithms can be physically tested.

After selecting the type of actuator and spatial configuration, the remaining part of the simulator design is choosing values of predefined geometrical parameters that fully define the mechanism with all constraints. Due to many different combinations and their interrelated influence on the simulator performance, it is challenging to find optimal values for these parameters. In this case, with many parameters, often some optimization algorithms must be employed to get sufficiently good results in the desired time. For problems like this, the optimization algorithms are often numerical and do not provide the solution in the closed form. These algorithms are traditionally iterative. Constrained minimization problems can be solved with minimization of the fitness function by successive approaching within a sequence of calculations. There are already reasonable efforts in employing the genetic algorithm [28–30] for optimization of this kind of problem. It is suitable because there are many local minima, and it is not easy to find a global one. Brute force and exhaustive algorithms are not chosen in this case because of the computational power and time necessary for optimization. There are also more recent evolutionary and bio-inspired metaheuristic algorithms that can be used for this kind of problem, such as the ones described in [31–33].

The implementation of the fitness function defines the goal of optimization. Many performance indicators can be considered in this case while calculating fitness functions, such as the maximum feasible orientation of the platform, the volume of workspace, needed joint movements, manipulability, dexterity, singularities, and physical interference between segments. This paper proposes novel constraints, performance indica-

tors, and fitness function for flight simulation on a low-cost flight simulator with electric rotary actuators. While constructing fitness function and choosing performance indicators, the goal was to lower the necessary computation time and to be able to find geometry with sufficient characteristics in a short time. On the other hand, the purpose of constraints is to make sure that obtained geometry can be materialized, that it is really suitable for flight simulation, and that there is no physical limitation on the materialized mechanism. Previously, inverse and forward kinematics problems had to be solved many times in this kind of optimization, the goal of the proposed algorithm is to greatly reduce the number of these calculations within optimization.

## 2. Geometric Parameters

In order to analyze whether the shown type of mechanism can be successfully applied in practice for the flight simulator, foremost, the geometry must be defined. The geometry should be defined with a minimum number of parameters that fully define the mechanism with all constraints. Changing the values of these parameters within the optimization process should provide better characteristics for the flight simulator.

As shown in Figure 3, the first step is to define two reference frames. A global reference frame ($O_b x_b y_b z_b$) is fixed to the center of the base and a moving platform reference frame ($O_p x_p y_p z_p$) is fixed to the center of the platform.

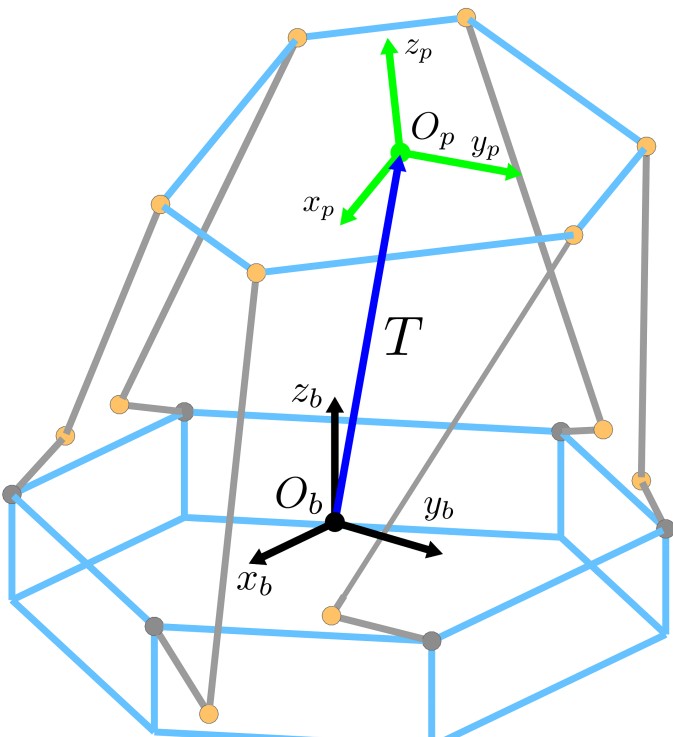

**Figure 3.** Global and platform's reference frame.

Control values are defined by vector $\boldsymbol{x} = \begin{bmatrix} x & y & z & \psi & \theta & \varphi \end{bmatrix}^T$, and they represent external coordinates of the moving platform's local reference frame in reference to the global frame fixed to the nonmoving base of the platform. In other words, translation of the platform's frame in reference to the global frame is defined by $\boldsymbol{T} = \begin{bmatrix} x_{b,O_p} & y_{b,O_p} & z_{b,O_p} \end{bmatrix}^T$, while orientation is defined using Euler angles ($\psi, \theta, \varphi$), with yaw, pitch, and roll about for $z$, $y$, and $x$ global axes, respectively.

The rotary actuator is fixed to the base of the mechanism, and it rotates the lower lever. Figure 4 shows point $B_i$; this is the point of connection between the rotary actuator and the lower lever, which means that it is the intersection point of the actuator's axis of rotation

(defined with $\boldsymbol{i_{m,i}}$) and the lower lever. The lower lever has a constant length (from point $B_i$ to point $A_i$), and it is equal to parameter $a$, while the constant length of the upper lever that connects points $B_i$ and $P_i$ is equal to parameter $s$. These are the first two geometric parameters. In reference to the horizontal plane on which the base lies, the actuator's axis of rotation is tilted by the $\varepsilon$ angle, the third geometric parameter.

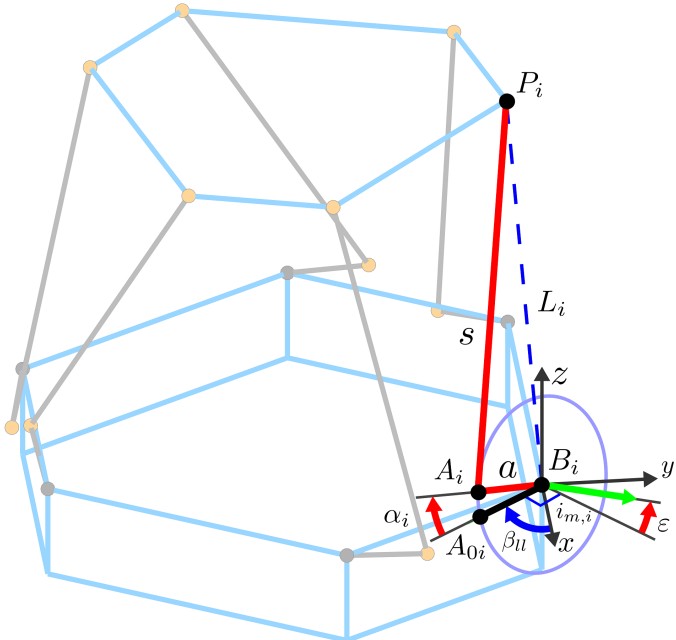

**Figure 4.** Geometric parameters of one leg.

Point $P_i$ is a point on the platform around which the spherical joint rotates, the sphere's center. Point $A_i$ is the center of the spherical joint connecting the upper and lower lever. Due to the six legs of this mechanism $i = 1, 2, \dots, 6$.

The vector $\boldsymbol{p_i} = \begin{bmatrix} x_{Pi} & y_{Pi} & z_{Pi} \end{bmatrix}^T$ is the position vector of a point $P_i$ in the reference frame of the platform. A vector $\boldsymbol{b_i} = \begin{bmatrix} x_{Bi} & y_{Bi} & z_{Bi} \end{bmatrix}^T$ is the position vector of a point $B_i$ in the global reference frame.

Coordinates of points $B_i$ and $P_i$ must be determined in corresponding reference frames in order to fully define the mechanism's base and platform. The intuitive way to do this is to use polar coordinates, noticing that points $B_i$ and $P_i$ lie on constant diameter circles and that there are three axes of symmetry between these points (point $B_1$ can be mapped to point $B_2$, while the axis of symmetry is at a certain angle to the $x$ axis of global frame).

As shown in Figure 5, the axis of symmetry between points $B_1$ and $B_2$ is defined by angle $\beta_{s,i}$. This angle has a constant value for all considered geometries, and its purpose is to define three segments for points $B_i$. The axis of symmetry of points $B_1$ and $B_2$ is rotated by $30°$ in reference to the $x$-axis, then the axis of symmetry of points $B_3$ and $B_4$ is rotated by $150°$, and the axis of symmetry of points $B_5$ and $B_6$ is rotated by $-90°$. These three axes of symmetry make the distribution of points uniform. Angle $\beta_{s,i}$ is not a geometrical parameter for optimization, but it is necessary for determining the coordinates of points $B_i$ with other parameters. The value of this angle for all legs is defined with vector $\boldsymbol{\beta_s} = \begin{bmatrix} \frac{1}{6}\pi & \frac{1}{6}\pi & \frac{5}{6}\pi & \frac{5}{6}\pi & -\frac{1}{2}\pi & -\frac{1}{2}\pi \end{bmatrix}^T$.

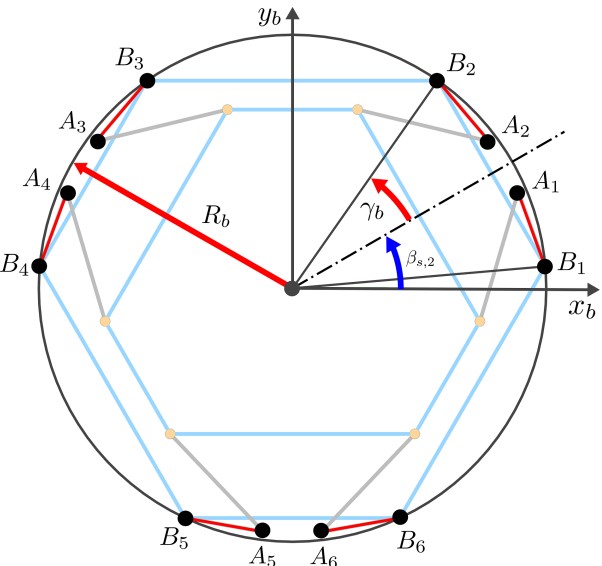

**Figure 5.** Geometric parameters of the mechanism's base.

Angle $\gamma_b$ and radius $R_b$ are the next two variable geometric parameters that are used to define the position of the point $B_i$ relative to the previously defined axis of symmetry. In order to define both points $B_1$ and $B_2$, located on different sides relative to the axis of symmetry, a vector $\sigma_l = \begin{bmatrix} -1 & 1 & -1 & 1 & -1 & 1 \end{bmatrix}$ is introduced. This vector also has a constant value for all considered geometries. Its purpose is also to define different rotation directions of even and odd indexed actuators.

Finally, the position of all six points $B_i$ is defined as follows:

$$x_{B,i} = R_b \cos(\delta_{b,i}), \quad y_{B,i} = R_b \sin(\delta_{b,i}), \quad z_{B,i} = 0, \quad \boldsymbol{\delta_b} = \boldsymbol{\beta_s} + \gamma_b \, \boldsymbol{\sigma_l} \tag{1}$$

Similar to points $B_i$, the position of all six points $P_i$ in reference to the platform's reference frame is defined by the following two variable geometric parameters, radius $R_p$ and angle $\gamma_p$, as shown in Figure 6. There are again three axes of symmetry between points, the same as in the case of the base.

$$x_{p,i} = R_p \cos(\delta_{p,i}), \quad y_{p,i} = R_p \sin(\delta_{p,i}), \quad z_{p,i} = 0, \quad \boldsymbol{\delta_p} = \boldsymbol{\beta_s} + \gamma_p \, \boldsymbol{\sigma_l} \tag{2}$$

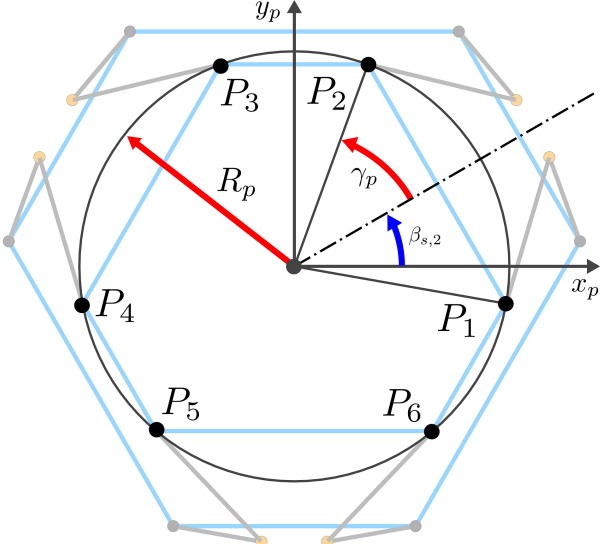

**Figure 6.** Geometric parameters of the mechanism's platform.

It is now necessary to define the position of points $A_i$, which depends on the previously mentioned vector, $i_{m,i}$ which is the unit vector of the actuator's axis of rotation.

As shown in Figure 7, the actuator's axis of rotation does not have to be normal to the line passing through points $B_1$ and $B_2$ but this line is a referent line for the definition of the orientation of lower levers in the horizontal plane (when lower levers are in the home position). This referent line is defined with the angle $v_{ll,i}$ measured from the $x$ axis of the global reference frame. The value of this angle is defined with a vector $\boldsymbol{v}_{ll} = \boldsymbol{\beta}_s - \frac{\pi}{2}\boldsymbol{\sigma}_l$ for each lower lever.

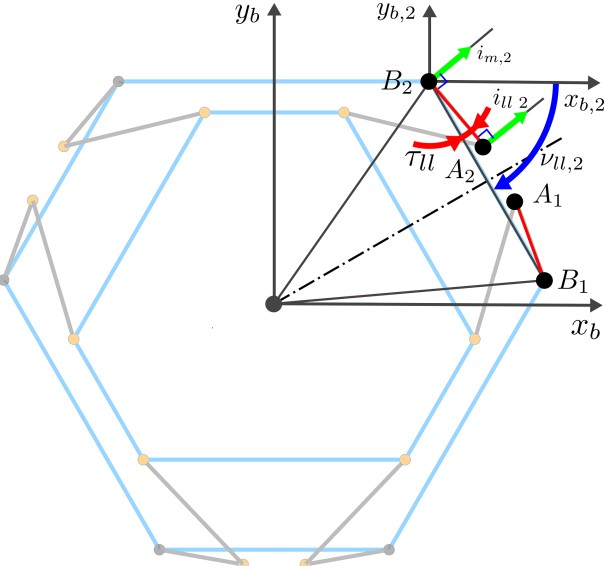

**Figure 7.** Axes of actuator and lower lever joint.

Finally, the angle between the lower lever's referent line and the actual orientation of the lower lever in the home position is the next geometric parameter $\tau_{ll}$, which is the same for all six lower levers. Then, the angle between the lower lever and the $x_b$ axis is given by the following equation:

$$\boldsymbol{\beta}_{ll} = \boldsymbol{v}_{ll} + \tau_{ll}\boldsymbol{\sigma}_l \tag{3}$$

The projection of the unit vector of the actuator's axis of rotation in the horizontal plane is normal to the lower lever. Based on that and the already defined tilt angle $\varepsilon$, the unit vector of the actuator's axis of rotation is:

$$\boldsymbol{i}_{m,i} = \begin{bmatrix} \cos\left(\beta_{ll,i} + \frac{\pi}{2}\sigma_{l,i}\right) \cos\varepsilon \\ \sin\left(\beta_{ll,i} + \frac{\pi}{2}\sigma_{l,i}\right) \cos\varepsilon \\ \sin\varepsilon \end{bmatrix} \tag{4}$$

Point $A_{0i}$ (shown in Figure 4) is the referent point that represents a point $A_i$ in the home position and its position vector $\boldsymbol{a_{0i}} = \begin{bmatrix} x_{A_{0i}} & y_{A_{0i}} & z_{A_{0i}} \end{bmatrix}^T$ is defined with:

$$x_{A_{0i}} = a\cos\beta_{ll,i} + x_{B,i}, \quad y_{A_{0i}} = a\sin\beta_{ll,i} + y_{B,i}, \quad z_{A_{0i}} = z_{B,i} \tag{5}$$

To define a point $A_i$ and position vector $\boldsymbol{a_i} = \begin{bmatrix} x_{Ai} & y_{Ai} & z_{Ai} \end{bmatrix}^T$ in the global reference frame, it is first necessary to define or find angle $\alpha_i$ (also shown in Figure 4). Then, based on Rodrigues' rotation formula, coordinates of the vector $\boldsymbol{a_i}$ can be expressed as follows:

$$\boldsymbol{a_i} = \boldsymbol{b_i} + \boldsymbol{B_i A_{0i}} + (1 - \cos\alpha_i)\, \boldsymbol{Rm1}_{B_i A_i} + (\sin\alpha_i)\, \boldsymbol{Rm2}_{B_i A_i}, \tag{6}$$

where $\boldsymbol{Rm2}_{B_i A_i}$ and $\boldsymbol{Rm1}_{B_i A_i}$ are:

$$\boldsymbol{Rm2}_{B_i A_i} = \boldsymbol{i}_{m,i} \times \boldsymbol{B_i A_{0i}}, \quad \boldsymbol{Rm1}_{B_i A_i} = \boldsymbol{i}_{m,i} \times \boldsymbol{Rm2}_{BA} \tag{7}$$

With all previously defined points, the geometry of the mechanism is fully defined. Still, it is necessary to define two more unit vectors for each leg in order to be able to model the constraints of spherical joints. These unit vectors of spherical joints are essential for optimization, as their orientation is used for the calculation of the physical limitations of this mechanism. This physical limitation is modeled as a nonlinear constraint because at a certain maximal angle of the joint, there is contact between different parts of the joint, and further movement is denied. It is expected that the optimization algorithm adapts the orientation of these vectors in order to get the largest possible workspace.

Figure 8 shows a projection of the unit vector of the upper joint axis $i_{jp,i}$ in the plane defined by $x_p$ and $y_p$. It is also shown that angle $\tau_{jp}$ is used to define its orientation. This angle is the next geometric parameter. The orientation of the unit vector of the upper joint axis in this plane (angle between vector projection and $x_p$ axis) is defined as follows:

$$\boldsymbol{\beta_{jp}} = \boldsymbol{\beta_s} + \tau_{jp}\,\boldsymbol{\sigma_l} \tag{8}$$

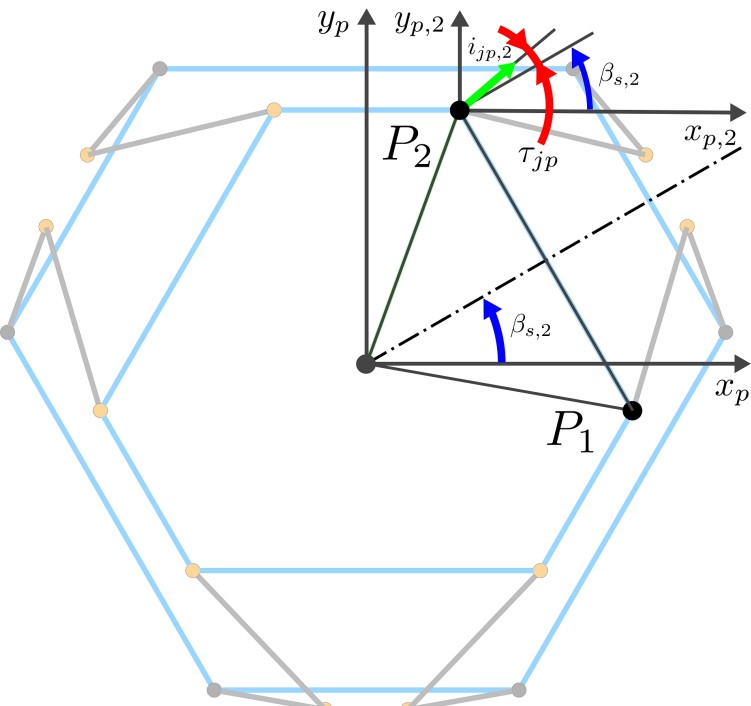

**Figure 8.** Axes of platform and upper lever joint.

A previously analyzed unit vector is used for the calculation of the physical limitation of the spherical joint that connects the platform and upper lever. The last unit vector is $i_{ll}$, which defines the axis of the spherical joint that connects the upper and lower lever. This unit vector is already shown in Figure 7. In the home position, its projection to the horizontal plane has the same direction as the projection of the actuator's axis of rotation.

Unit vectors of joints are again shown in Figure 9; their orientation in planes of the platform and the base are already discussed, but two more and two last geometric parameters are introduced for total orientation. Vectors are tilted in reference to the corresponding planes by angles $\mu_{jp}$ and $\mu_{ll}$. The same principle is already used for the actuators, and the following equation is obtained:

$$\boldsymbol{i_{jp,i}} = \begin{bmatrix} \cos\beta_{jp,i}\cos\mu_{jp} & \sin\beta_{jp,i}\cos\mu_{jp} & \sin\mu_{jp} \end{bmatrix}^T \tag{9}$$

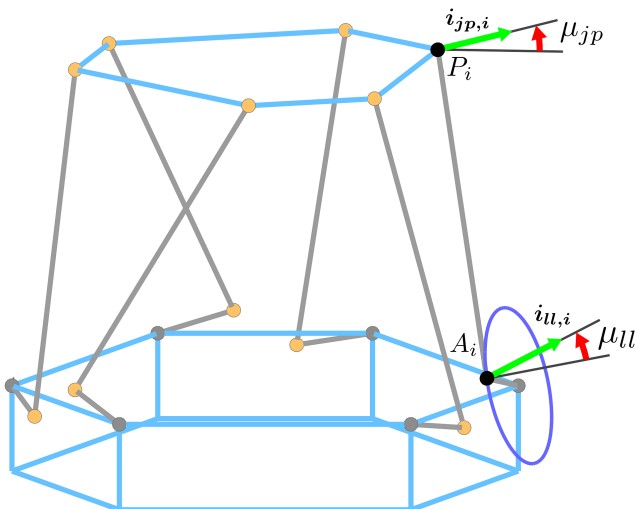

**Figure 9.** Axes of spherical joints.

Because one part of the joint between the lower and upper lever is fixed to the lower lever, it rotates together with the lower lever, and it is necessary to define the referent vector of the axis in home position $i_{ll0,i}$ with:

$$i_{ll0,i} = \begin{bmatrix} \cos\left(\beta_{ll,i} + \frac{\pi}{2}\sigma_{l,i}\right)\,\cos(\varepsilon + \mu_{ll}) \\ \sin\left(\beta_{ll,i} + \frac{\pi}{2}\sigma_{l,i}\right)\,\cos(\varepsilon + \mu_{ll}) \\ \sin(\varepsilon + \mu_{ll}) \end{bmatrix} \tag{10}$$

Then, again based on Rodrigues' rotation formula, the vector $i_{ll,i}$ can be expressed as follows:

$$i_{ll,i} = i_{ll0,i} + (1 - \cos\alpha_i)\,Rm1_{i_{ll}} + (\sin\alpha_i)\,Rm2_{i_{ll}}, \tag{11}$$

where $Rm2_{i_{ll}}$ and $Rm1_{i_{ll}}$ are:

$$Rm2_{i_{ll}} = i_{m,i} \times i_{ll0,i}, \quad Rm1_{i_{ll}} = i_{m,i} \times Rm2_{i_{ll}} \tag{12}$$

To optimize the geometry, eleven geometric parameters that determine the geometry of the mechanism for flight simulators are precisely defined. They unambiguously describe platform geometry. It should be noted that the geometry could be defined in other ways. Still, because no restrictions are imposed besides symmetry between joints (mirroring of, i.e., point $B_1$ to point $B_2$), any usable configuration can be obtained. It is necessary only once to implement the previously described algorithm for calculating the position and axes vectors based on geometric parameters. Then it is as simple as changing 11 numbers to obtain different mechanism geometry. One example of values for all geometric parameters is given in Table 1.

**Table 1.** Example of geometric parameter values for some mechanism.

| Parameter | Value |
|:---:|:---:|
| $a$ | 230 mm |
| $s$ | 1110 mm |
| $R_p$ | 520 mm |
| $R_b$ | 840 mm |
| $\gamma_p$ | 7° |
| $\gamma_b$ | 35° |
| $\varepsilon$ | −39° |
| $\tau_{ll}$ | −122° |
| $\tau_{jp}$ | −12° |
| $\mu_{ll}$ | −18° |
| $\mu_{jp}$ | −1° |

### 3. Inverse and Forward Kinematics Problems

In kinematics, joint movements are referred to as generalized or internal coordinates, and external coordinates are the ones that define the position and orientation of segments in reference to a global reference frame. Changing internal coordinates changes external coordinates, i.e., the position and orientation of the platform. The process of determining external coordinates for a given set of internal coordinates (joint movements) is referred to as forward kinematics. Determining internal coordinates (joint movements) for a given set of external coordinates (position and orientation) is referred to as inverse kinematics.

#### 3.1. Coordinate System Transformation

Besides the translation vector $T$ of the platform's frame in reference to the global frame, it is necessary also to define a matrix $R$ that represents the rotation matrix, whose elements are functions of the three Euler angles that determine the orientation of the moving platform's reference frame in reference to the global frame. To ease writing, a condensed notation of the trigonometric functions sine and the cosine, "$s$" and "$c$", respectively, are used to define the rotation matrix $R$ as follows:

$$R = R_z(\psi)R_y(\theta)R_x(\varphi) = \begin{bmatrix} c_\psi\,c_\theta & -s_\psi\,c_\varphi + c_\psi\,s_\theta\,s_\varphi & s_\psi\,s_\varphi + c_\psi\,s_\theta\,c_\varphi \\ s_\psi\,c_\theta & c_\psi\,c_\varphi + s_\psi\,s_\theta\,s_\varphi & -c_\psi\,s_\varphi + s_\psi\,s_\theta\,c_\varphi \\ -s_\theta & c_\theta\,s_\varphi & c_\theta\,c_\varphi \end{bmatrix} \tag{13}$$

#### 3.2. Inverse Kinematics

Calculating the required rotations of rotary actuators and lower levers for a given position and orientation of the platform requires solving the inverse kinematics. The most intuitive way of solving this type of problem is often referred to as a geometric method. Using translation vector $T$ and rotation matrix $R$, it is possible to obtain a position vector of a point $P_i$ in reference to the global reference frame.

The necessary length of the leg (distance between points $B_i$ and $P_i$) can be computed by the Euclidean norm of the vector $L_i$ shown in Figure 4 as $l_i = \|L_i\|_2$. After calculating the vector $L_i$, the next step is to find the angle of rotation $\alpha_i$ for the rotary actuator. This angle defines rotation around the axis of the actuator, while this axis is defined by a unit vector $i_{m,i}$. The purpose of finding the angle $\alpha_i$ is to define the position of the point $A_i$, which provides the correct position of upper and lower levers for the defined leg length. Angle $\alpha_i$ is measured from the initial vector $B_iA_i$, which lies in the horizontal plane. The actuator's axis of rotation and the circle representing all possible positions of the point $A_i$ are also shown in Figure 4. For the introduced reference frames, the following relation determines the vector of the $i$-th leg.

$$L_i = T + R\,p_i - b_i \tag{14}$$

Home position offset can be implemented in the already defined translation vector $T$ as follows:

$$T = \begin{bmatrix} x & y & z_0 + z \end{bmatrix}^T \tag{15}$$

The home position is the selected position in which lower levers are in plane with the base. This position of the platform is defined as:

$$z_0 = \sqrt{s^2 - (x_{P_1} - x_{A_1})^2 - (y_{P_1} - y_{A_1})^2} + z_{A_1} \tag{16}$$

Using now known vectors $p_i$ and $b_i$, and known lengths $a$, $s$, and $L_i$, angle $\alpha_i$ can be obtained from coordinates of vector $a_i$. After solving a system of equations and using trigonometric identities, the following equation is obtained:

$$\alpha = \sin^{-1}\left(\frac{D}{\sqrt{E^2 + F^2}}\right) - \text{atan2}(F, E) \tag{17}$$

The function atan2 from the previous equation is defined as:

$$
\text{atan2}(F_i, E_i) = \begin{cases} \tan^{-1}\left(\dfrac{F_i}{E_i}\right), & E_i > 0, \\[2mm] \tan^{-1}\left(\dfrac{F_i}{E_i}\right) + \pi, & E_i < 0 \wedge F_i \geq 0, \\[2mm] \tan^{-1}\left(\dfrac{F_i}{E_i}\right) - \pi, & E_i < 0 \wedge F_i < 0, \\[2mm] +\dfrac{\pi}{2}, & E_i = 0 \wedge F_i > 0, \\[2mm] -\dfrac{\pi}{2}, & E_i = 0 \wedge F_i < 0, \\[2mm] \text{NaN}, & E_i = 0 \wedge F_i = 0. \end{cases} \tag{18}
$$

In Equation (17), variables **D**, **E**, and **F** are:

$$
D = l^2 + a^2 - s^2 - 2(Rm\mathbf{1}_{BA} + BA_0)\,BP \tag{19}
$$

$$
E = 2Rm\mathbf{2}_{BA} \cdot BP \tag{20}
$$

$$
F = -2Rm\mathbf{1}_{BA} \cdot BP \tag{21}
$$

With the obtained Equation (17), the inverse kinematics problem is solved. This equation is the explicit solution to the inverse kinematics problem. If there is a real solution to this equation for all angles $\alpha_i$, then the desired position and orientation of the platform are achievable if there are no other physical constraints.

In the process of designing a mechanism like this for practical usage as a flight simulator, the physical constraints of the kinematic chains, such as lever interference and limitations of joints, must be considered [9]. Whenever some mechanism has spherical or universal joints, their angular limits should be considered [29]. The range of work has to be defined for each joint of the mechanism. The most commonly available spherical joint is called rod end bearing. The nominal position of levers for this joint is when levers are orthogonal to each other. Based on this, if the rotation between the ends of the upper lever is possible, the constraints of the upper and lower joints can be defined as follows:

$$
\lambda_{p,i} = \left| \cos^{-1}\left(\frac{P_i A_i \cdot i_{jpb,i}}{s}\right) - 90^\circ \right| < \lambda_{max}, \quad i_{jpb,i} = R\,i_{jp,i} \tag{22}
$$

$$
\lambda_{b,i} = \left| \cos^{-1}\left(\frac{P_i A_i \cdot i_{ll,i}}{s}\right) - 90^\circ \right| < \lambda_{max} \tag{23}
$$

The maximal cone angle of this type of joint is $\lambda_{max}$, and it needs to be determined in advance for a specific joint.

### 3.3. Forward Kinematics

In the case of the inverse kinematics problem, the external coordinates are known, but if it is necessary to determine the external coordinates based on the known lower lever orientation, it would require solving the forward kinematics problem. The forward kinematics of this system is very complex and a lot of effort has been invested in solving it analytically and numerically.

The Jacobian matrix (matrix of all functions' first-order partial derivatives) maps the change of internal coordinates to the change of external coordinates with respect to time [34]. Based on the principle of virtual work, the Jacobian matrix, as a relationship between joint velocities and the platform velocities is also a direct relation between the

necessary torque of actuators and applied load to the platform [35]. For each leg of the mechanism, the equation of the loop closure (shown in Figure 10) is given by:

$$O_b B_i + B_i A_i + A_i P_i = O_b O_p + O_p P_i \qquad (24)$$

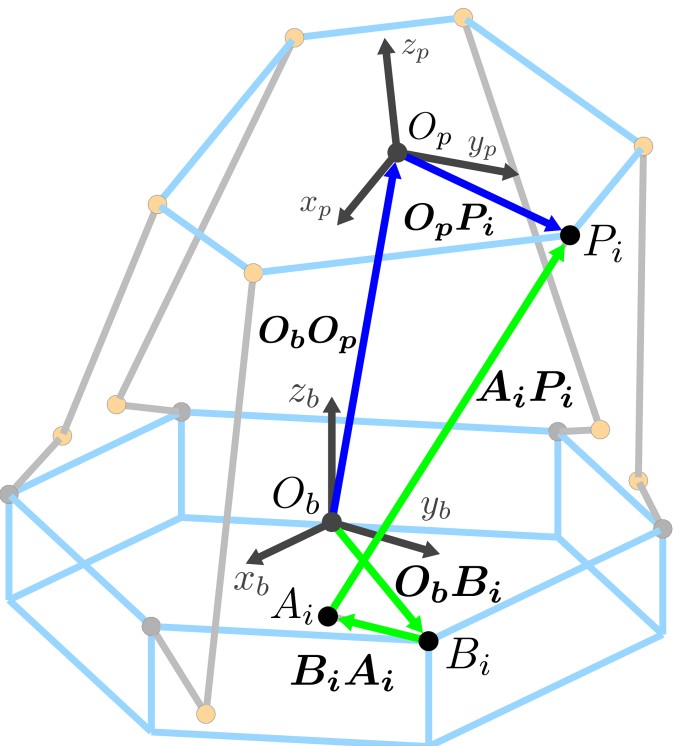

**Figure 10.** Single leg loop closure.

By differentiating Equation (24) with respect to time, the following equation is obtained:

$$\boldsymbol{\omega}_{\alpha,i} \times \boldsymbol{B}_i \boldsymbol{A}_i + \boldsymbol{\omega}_{ul,i} \times \boldsymbol{A}_i \boldsymbol{P}_i = \boldsymbol{v}_p + \boldsymbol{\omega}_p \times \boldsymbol{O}_p \boldsymbol{P}_i, \qquad (25)$$

where $\boldsymbol{v}_p$ is platform linear velocity, $\boldsymbol{\omega}_{\alpha,i}$, $\boldsymbol{\omega}_{ul,i}$, and $\boldsymbol{\omega}_p$ are angular velocities of the actuator (and lower lever), upper lever, and platform, respectively. By introducing the unit vector of the upper lever as $\boldsymbol{A}_i \boldsymbol{P}_i = s\boldsymbol{i}_{ul,i}$, multiplying both sides of the equality with it, and using characteristics of the cross product, the equation becomes:

$$\boldsymbol{i}_{ul,i}(\boldsymbol{\omega}_{\alpha,i} \times \boldsymbol{B}_i \boldsymbol{A}_i) + \boldsymbol{\omega}_{ul,i}(s\boldsymbol{i}_{ul,i} \times \boldsymbol{i}_{ul,i}) = \boldsymbol{i}_{ul,i} \boldsymbol{v}_p + (\boldsymbol{O}_p \boldsymbol{P}_i \times \boldsymbol{i}_{ul,i})\boldsymbol{\omega}_p \qquad (26)$$

By introducing the unit vector of rotary actuator $\boldsymbol{i}_{m,i}$ and his angular velocity $\dot{\alpha}_i$ with $\boldsymbol{\omega}_{\alpha,i} = -\sigma_{l,i}\dot{\alpha}_i \boldsymbol{i}_{m,i}$ and by eliminating the second member of the equation, the following equation is obtained:

$$-\sigma_{l,i}\boldsymbol{i}_{ul,i}(\boldsymbol{i}_{m,i} \times \boldsymbol{B}_i \boldsymbol{A}_i)\dot{\alpha}_i = \boldsymbol{i}_{ul,i} \boldsymbol{v}_p + (\boldsymbol{O}_p \boldsymbol{P}_i \times \boldsymbol{i}_{ul,i})\boldsymbol{\omega}_p \qquad (27)$$

Based on the previous equation, the Jacobian matrix of external coordinates $\boldsymbol{J}_x$ and internal coordinates $\boldsymbol{J}_\alpha$ can be defined as follows:

$$\boldsymbol{J}_\alpha \dot{\boldsymbol{\alpha}} = \boldsymbol{J}_x \dot{\boldsymbol{x}} \qquad (28)$$

$$\boldsymbol{J}_x = \begin{bmatrix} \boldsymbol{i}_{ul,1} & \boldsymbol{O}_p \boldsymbol{P}_1 \times \boldsymbol{i}_{ul,1} \\ \vdots & \vdots \\ \boldsymbol{i}_{ul,6} & \boldsymbol{O}_p \boldsymbol{P}_6 \times \boldsymbol{i}_{ul,6} \end{bmatrix} \qquad (29)$$

$$J_\alpha = \begin{bmatrix} -\sigma_{l,1}\, \boldsymbol{i_{ul,1}}(\boldsymbol{i_{m,1}} \times \boldsymbol{B_1 A_1}) & & \\ & \ddots & \\ & & -\sigma_{l,6}\, \boldsymbol{i_{ul,6}}(\boldsymbol{i_{m,6}} \times \boldsymbol{B_6 A_6}) \end{bmatrix} \tag{30}$$

$$\dot{\boldsymbol{x}} = \begin{bmatrix} v_{px} & v_{py} & v_{pz} & \omega_{px} & \omega_{py} & \omega_{pz} \end{bmatrix}^T, \quad \dot{\boldsymbol{\alpha}} = \begin{bmatrix} \dot\alpha_1 & \cdots & \dot\alpha_6 \end{bmatrix}^T \tag{31}$$

Finally, the Jacobian matrix $\boldsymbol{J}$ can be expressed with the following equation:

$$\boldsymbol{J} = \boldsymbol{J_x}^{-1} \boldsymbol{J_\alpha} \tag{32}$$

If $\boldsymbol{\alpha}^*$ is a vector of desired angles, then based on Newton's method, the solution of the forward kinematics problem would be:

$$\boldsymbol{x}^{k+1} = \boldsymbol{x}^k - \boldsymbol{J}\Big(\boldsymbol{\alpha}(\boldsymbol{x}^k) - \boldsymbol{\alpha}^*\Big) \tag{33}$$

In this way, the simple iterative numerical solution to the forward kinematics problem of the rotary Stewart platform is obtained. A good approximation (with an error smaller than the defined threshold [36]) of external coordinates is obtained from Equation (33) after just a few iterations.

## 4. Mechanism Performance Indicators

The first requirement in order to apply optimization is that there is a certain parameter on the basis of which it is possible to measure the success of individual solutions and then compare different solutions. It is necessary to construct a function whose value will be a measure of the success of the solution. This function is not always easy to form when it comes to complex engineering problems. It is necessary to determine the parameter that will represent the performance indicators, in this case, of the flight simulator mechanism.

### 4.1. Required Actuator Torque

Based on Jacobian matrix that is for this mechanism defined with Equation (32), it is possible to find a required torque of actuators for payload on the platform.

With the inverse of the matrix $\boldsymbol{J}$, and if the load acting upon the platform $\boldsymbol{F_x}$ is known, the required torque of actuators $\boldsymbol{M_m}$ can be found with:

$$\boldsymbol{M_m} = \boldsymbol{J}^T \boldsymbol{F_x} \tag{34}$$

The previous equation was obtained based on the principle of virtual work because:

$$\delta \boldsymbol{x} = \boldsymbol{J} \delta \boldsymbol{\alpha} \tag{35}$$

In case of static equilibrium, if $\boldsymbol{C_{in}}$ is the center of inertia of the payload, and if just the weight of the payload $(m\boldsymbol{g})$ is acting upon the platform, then the vector $\boldsymbol{F_x}$ can be written as:

$$\boldsymbol{F_x} = \begin{bmatrix} m\boldsymbol{g} & (\boldsymbol{C_{in}} - \boldsymbol{O_p}) \times m\boldsymbol{g} \end{bmatrix}^T \tag{36}$$

Required actuator torque for motion (dynamic case) can be defined using the static case and load factor or by dynamic analysis and simulation of the platform, as in [37].

### 4.2. Workspace

One of the most important characteristics of the mechanism used for flight simulation is its workspace. A good example is a flight simulator for fighter aircraft, as shown in [38], for which the Stewart platform is not suitable because of workspace limits. The workspace of any mechanism (including parallel mechanisms, which has a closed kinematic chain) is a set of positions and orientations reachable by its end-effector. In order to be able to successfully simulate flight with a motion platform, on which the pilot sits while in

training, its workspace has to meet some criteria. Based on the preceding, it is essential to have a computationally fast and efficient but at the same time also accurate workspace determination process in order to design and optimize the geometry of the mechanism used for the flight simulator.

There is no analytical solution for the workspace of this type of mechanism that considers all constraints (such as motion limits of joints), which can be practically used in the design process. One option is to simply test all significant positions and orientations. This includes defining the range for each axis and value change step (increment between consecutive values) and then testing all possible combinations. In order to lower the number of combinations that must be tested, space is divided a few times, first with the coarser step and then with the finer one just around the boundary of the workspace. The final step is defining the boundary surface based on points that are within the workspace.

The workspace of the mechanism is embedded in a six-dimensional space that cannot be graphically represented. Because of that, Figure 11 shows the workspace for constant orientation, in this case when all Euler angles are zero.

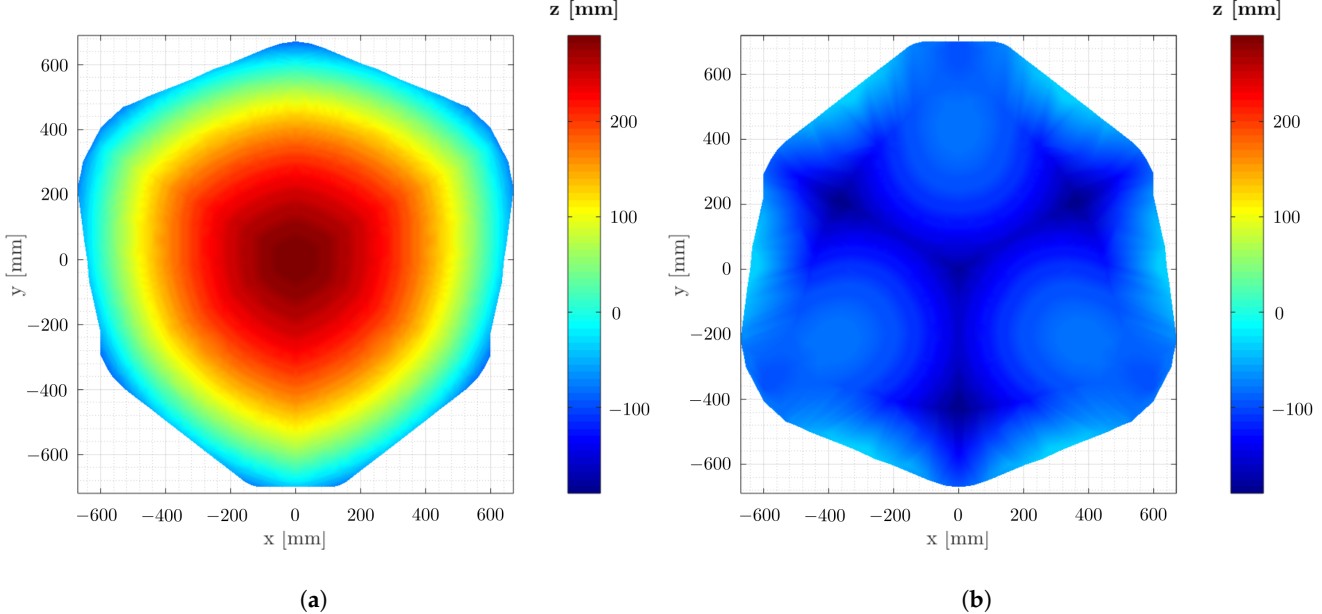

(**a**)　　　　　　　　　　　　　　　　　　　　　　　　　　　　(**b**)

**Figure 11.** (**a**) Top and (**b**) bottom view of the workspace for zero orientation.

When the workspace is not suitable for the desired motion, filters called washout algorithms are used to provide the feeling as if the real motion has been achieved. For flight simulators, motion cueing [39–41] and washout filters [42–44] are used in order to improve the capabilities of the mechanism itself, but the degree of improvement depends on the pilot. Paper [45] presents an emulation of pilot control behavior on a flight simulator to further improve control of the mechanism.

### 4.3. Other Performance Indicators

As previously mentioned, performance indicators must allow solutions to be compared and, for that reason, must be based on normalized quantities. From many performance indicators that can be considered while calculating fitness functions, dexterity and the maximum feasible Euler angles have been selected as the most valuable for flight simulation. In this case, it is necessary to normalize the Jacobian matrix because the elements have different units of measurement. Certain elements are divided by the platform's radius or by the length of the lower lever to perform normalization.

$$J_{xn} = \begin{bmatrix} i_{ul} & O_p P_i \times i_{ul}/R_p \end{bmatrix}, \; J_{\alpha n} = J_\alpha/a \tag{37}$$

Based on the previous two equations, the normalized Jacobian matrix is:

$$J_n = J_{xn}^{-1} J_{\alpha n} \tag{38}$$

In order to calculate the Local Dexterity Index (LDI) $\eta$ as the reciprocal value of the Jacobian matrix condition number [16], a homogeneous Jacobian matrix $J_h$ should first be obtained. As shown in [20], the Jacobian matrix of external coordinates $J_x$ can be separated into two matrices that have different physical dimensions. Then some sort of normalization can be applied to obtain a homogeneous matrix. The first matrix, obtained as the first three columns, is related to forces, while the last three columns are related to torques.

Paper [46] defines robot manipulability as the absolute value of the determinant of the Jacobian matrix, which in this case was previously normalized.

$$w = |\det(J_n)| \tag{39}$$

Paper [34] defines robot dexterity as the reciprocal value of the condition number of the Jacobian matrix, which in this case was previously normalized. The condition number of the matrix is the product of the Euclidean norm of the matrix and the Euclidean norm of the inverse matrix.

$$\eta = \frac{1}{k} = \frac{1}{\text{cond}(J_n)} = \frac{1}{\|J_n\|_2 \cdot \left\|J_n^{-1}\right\|_2} \tag{40}$$

Manipulability and dexterity of this type of mechanism are also analyzed in papers [47–49]. For some applications, performance indicators can also be partial or full isotropy of the mechanism described in [50].

The third and fourth performance indicators are based on the maximum absolute value of the feasible independent Euler pitch and roll angles ($\theta_{j,max}$ and $\varphi_{j,max}$) about global y- and x-axes for the given position. In relation to the method with the incremental change of angle with constant step, until the maximum value (with a certain tolerance) is determined, a significantly more efficient method is implemented. This new method is based on the variable step and determining the node in which the indicator of feasibility changes value, certain tolerance is met.

An additional indicator, for high-precision robots, is the elasticity of their structure. The mechanism can be designed to withstand all loads within some allowed limits of displacement, or this can be computed or even measured and then compensated with a control algorithm. In both cases, in the design process, displacement of the structure under load has to be calculated. The interesting part of this analysis is the numerical modeling of joints. The finite element method can be applied to obtain displacements of this mechanism, while joints can be implemented using the Lagrangian-multiplier method.

## 5. Real Coded Mixed Integer Genetic Algorithm

Genetic algorithm (GA) is one of the optimization and search techniques based on genetics and natural selection, modeled on the idea of Darwin's theory of species origin by means of natural selection. Using this technique, the problem is solved for a different set of inputs, and each set is given some score which is then used as a selection criterion. This set of inputs is called the population of one generation. Before solving the problem again, the set of inputs is changed using elitism, selection, crossover, and mutation functions. The score which is given to an individual solution is in fact the value of a fitness function for a given set of inputs. The idea is to provide an algorithm that evolves toward a globally optimal solution.

Figure 12 shows a flowchart of the real coded mixed integer genetic algorithm (RCMIGA) that is defined in [51]. This particular algorithm is selected because of the large number of inputs (in this case, eleven geometric parameters) and the highly nonlinear and discontinuous nature of the problem with many constraints and many local minima.

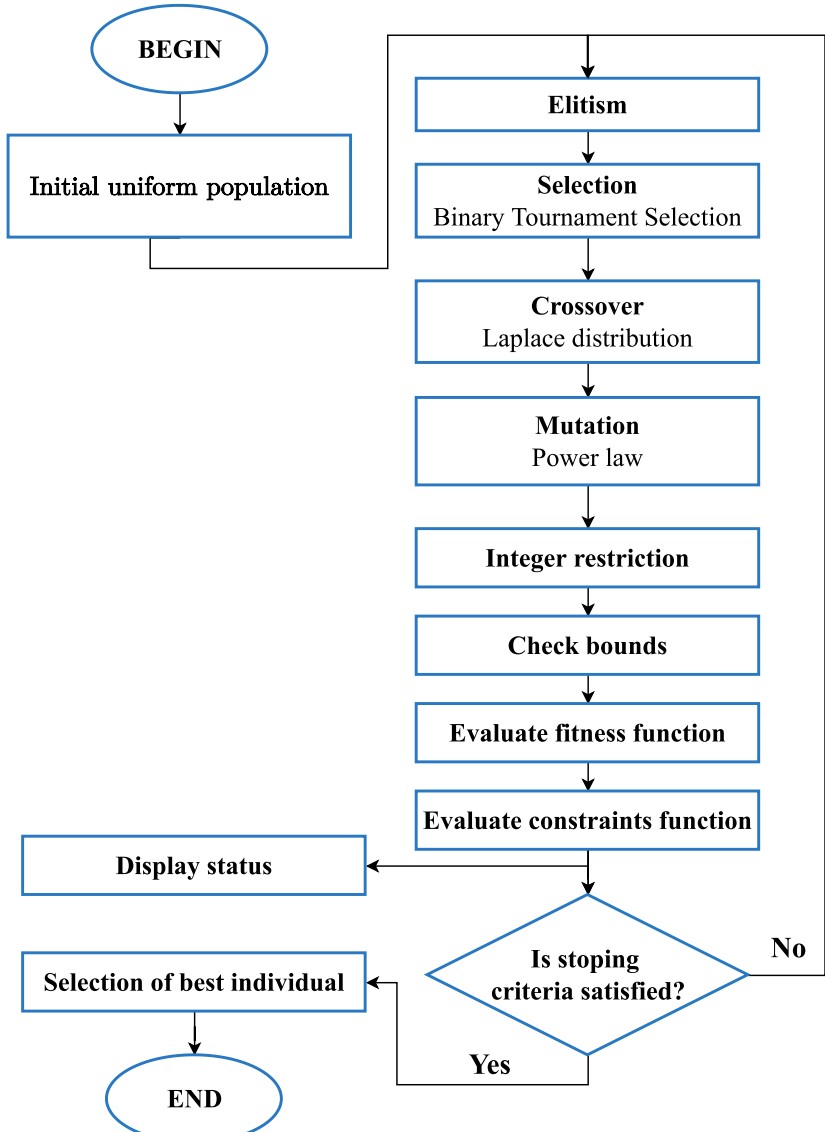

**Figure 12.** Flowchart of real coded mixed integer genetic algorithm.

In aerospace engineering, genetic algorithms are used even for the design of the whole aircraft, as in [52]. It is also chosen for washout filter tunning in [44].

This type of GA was chosen because the manufacturing technology requires that the geometric parameters are not continuous values, but there is some accuracy with which it is possible to manufacture parts. From this comes the explanation for the term "mixed integer" in the name, it is simply possible to define some variables so that they take only integer values. In addition, "real coded" describes that real numbers are used to represent individuals. Obtained integer values can be mapped to available discrete values, these discrete values can be real numbers. Values to which integers are mapped can be arbitrary, but they are more often linearly spaced.

As one real engineering problem is analyzed here, limitations inevitably narrow the scope of possible solutions. Due to the need to define limits, the penalty method defined in the paper [53] is used. This method is reflected in the evaluation of nonlinear conditions and by summing the values of nonlinear conditions with the fitness function value. It should be noted that the value for the penalty based on the evaluation of the constraint function is obtained by normalizing the value of the constraint function.

Elitism has been implemented to ensure that the best individuals pass from generation to generation unchanged. Despite the fact that they remain unchanged, these individuals,

together with others, participate in the selection process so that they can be selected for crossover and mutation, i.e., the application of genetic operators over them.

In order to enter the optimization process at all, the first generation of the population is generated based on a uniform distribution. As for every subsequent generation, this initial population must have the parameter values within the defined limits, and if there are integer parameters, they are required to have an integer value.

When there is an initial population, it is possible to start the optimization process. First, the elite individuals are separated, and the selection is made. In this case, a selection based on a tournament between two individuals is used.

The genetic crossover operator is then applied to the selected individuals, a crossover based on the Laplace distribution. This crossover is defined in the paper [54]. This method has three parameters for adjustment and their values are taken from the same paper.

$$a = 0, \; b_{real} = 0.15, \; b_{int} = 0.35 \tag{41}$$

Then the genetic operator of the mutation is applied, and this mutation is based on the power distribution law. This mutation is defined in the paper [55]. This method has two parameters for adjustment, and their values are taken from the same paper.

$$p_{real} = 10, \; p_{int} = 4 \tag{42}$$

Because the optimization process can take a long time, functions have been implemented to show the optimization status so that data is displayed at the end of each generation's evaluation. Based on this data, a conclusion can be made about the course of that optimization.

Once all genetic operators have been implemented, and the evaluation of constraint and fitness functions are enabled, it remains to define stopping criteria for process termination. A total of seven stopping criteria have been implemented, and if any of these criteria are met, the loop optimization interrupt occurs.

## 6. Geometry Optimization

After defining all necessary geometric parameters, after adopting all the required vectors to describe the mechanism, and when the genetic algorithm is fully implemented, it remains to define constraints and fitness function as well as to determine values for all optimization settings.

The optimum design of Stewart platform for particular applications is analyzed in papers [56–58]. Their goals are maximum rigidity over the workspace, improved dexterity, sufficient dexterity, and regular target workspace while minimizing the lengths of the elements, minimizing energy consumption, and the tracking error of a target trajectory.

Besides that, paper [59] presents an optimization method for obtaining the desired workspace, while papers [60,61] are focused on Stewart platform optimization for flight simulators.

### 6.1. Constraints

An engineering design problem, such as this one, often has some constraints such as, for example, manufacturing limitations. For the optimization process, these limitations must be, in some way, taken into account.

The first optimization constraints are the upper and lower bounds of the proposed geometric parameters. It is easier to find a solution for a smaller search space. For this specific problem, the boundaries could be determined based on allowable overall dimensions of the flight simulator and dimensions of the payload. Even when bounds are known, often there is a standard set of values to choose from. This type of problem is known as a mixed integer problem. Table 2 shows predefined ranges of geometric parameter values and possible steps for values between bounds. If the precision of the manufacturing process is 0.01 mm, then the step can be 0.01 mm. If a larger step is chosen, the search space is smaller and if a

smaller step is chosen, it is not possible to produce obtained geometry with that precision. These values for steps are chosen in order to lower search space. In this way, search space and the standard set for values of geometric parameters are defined.

**Table 2.** Value range of geometric parameters for optimization.

| Parameter | Value Range | Step |
|:---:|:---:|:---:|
| $a$ | 100 mm to 400 mm | 10 mm |
| $s$ | 600 mm to 1000 mm | 10 mm |
| $R_p$ | 400 mm to 800 mm | 10 mm |
| $R_b$ | 700 mm to 1500 mm | 10 mm |
| $\gamma_p$ | 5° to 55° | 1° |
| $\gamma_b$ | 5° to 55° | 1° |
| $\varepsilon$ | −90° to 90° | 1° |
| $\tau_{ll}$ | −180° to 180° | 1° |
| $\tau_{jp}$ | −90° to 90° | 1° |
| $\mu_{ll}$ | −90° to 90° | 1° |
| $\mu_{jp}$ | −90° to 90° | 1° |

The next step is the definition of nonlinear constraints; two constraints of this type are given with Equations (22) and (23). If these inequalities are satisfied, then the desired position and orientation of the platform are achievable in terms of joints motion. In addition, there must be no interference between the legs of the mechanism. This type of constraint for lower levers can be defined as follows:

$$\min\left(\left\| \boldsymbol{D_i D_j} \right\|_2\right) > r_{min}, \; i \neq j, \tag{43}$$

where $D_i$ and $D_j$ are any points on closed line segments (from $B_i$ to $A_i$ and from $B_j$ to $A_j$, respectively) of two lower levers that are part of different legs. This ensures that the minimum distance (expressed with Euclidean norm) between any two lower levers is always larger than the predefined value $r_{min}$, and this can be adapted for any part of a mechanism.

The following constraints ensure that the selected actuator can be used. Nonlinear constraints based on necessary torque and allowed forces for actuators are given with the following inequality:

$$|\boldsymbol{M_{m,i}}| < M_{m,max}, \quad |\boldsymbol{Fr_R}| < Fr_{R,max}, \quad |\boldsymbol{Fa_R}| < Fa_{R,max}, \tag{44}$$

where $M_{m,max}$ is maximum allowable torque while $Fr_{R,max}$ and $Fa_{R,max}$ are maximal allowable radial and axial force for gearbox.

*6.2. Fitness Function*

If all defined constraints are satisfied for one set of geometric parameter values, then for this set, the corresponding score in the fitness function must be calculated. In this case, the implementation of the fitness function and chosen performance indicators need to ensure that the final solution is optimal for the flight simulator. It should be noted that the implementation of the fitness function should be tailored to a specific aircraft or class of aircraft.

To determine the value of the performance indicators, 27 points in space were selected (shown in Figure 13). These points are obtained as a mesh grid constructed with the following three vectors (dimensions are in mm):

$$\boldsymbol{x_g} = \begin{bmatrix} -150, 0, 150 \end{bmatrix}^T, \; \boldsymbol{y_g} = \begin{bmatrix} -150, 0, 150 \end{bmatrix}^T, \; \boldsymbol{z_g} = \begin{bmatrix} -75, 0, 75 \end{bmatrix}^T \tag{45}$$

Values of dexterity, maximal pitch, and roll angles as performance indicators ($\eta_j$, $\theta_{j,max}$, and $\varphi_{j,max}$, where $j = 1, 2, \ldots, 27$) are combined within a fitness function using a weighted sum approach (defined with the following equation). In addition, the boundaries for angles are introduced to obtain values between 0 and 1, while the value of $\eta_j$ is already in

this interval. The largest possible angles are calculated within predefined boundaries and with all constraints for each point.

$$y_{of} = 100 - \sum_{k=1}^{3} \frac{100}{27} \sum_{j=1}^{27} b_k I_{k,j}, \quad \sum_{k=1}^{3} b_k = 1 \tag{46}$$

After introducing performance indicators, the previous equation for calculation of penalty value (score) within the fitness function becomes:

$$y_{of} = 100 - \frac{100}{27} \sum_{j=1}^{27} \left( b_1 \eta_j + b_2 \frac{\theta_{j,max} - \theta_{j,min}}{2\theta_{max}} + b_3 \frac{\varphi_{j,max} - \varphi_{j,min}}{2\varphi_{max}} \right) \tag{47}$$

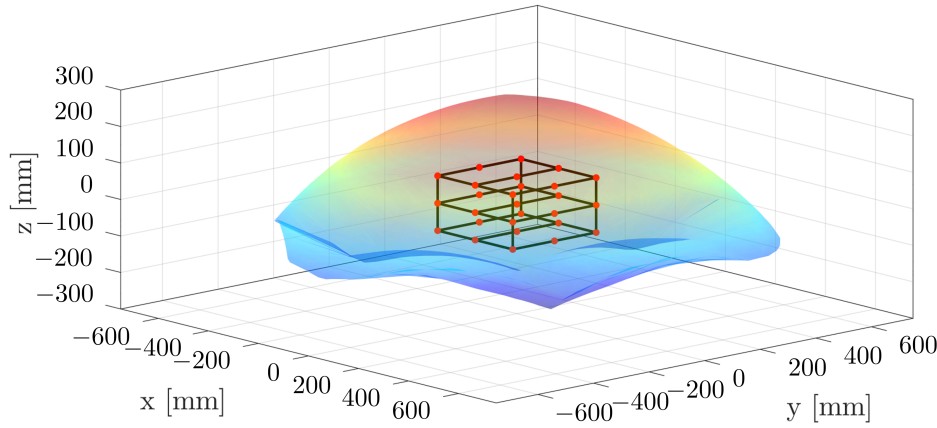

**Figure 13.** Red points are for evaluation of the performance indicators.

### 6.3. Settings

Table 3 shows the values of variables introduced in previous equations and which are necessary for the optimization process. These values are selected based on available electric actuators and gearboxes on the market, payload weight and size, and based on desired values of possible angles.

**Table 3.** Necessary optimization variables and their values.

| Variable | Value |
|---|---|
| Vector of sum weights | $\boldsymbol{b} = \begin{bmatrix} 0.4 & 0.3 & 0.3 \end{bmatrix}^T$ |
| Boundaries for angles | $\theta_{max} = 35°$, $\varphi_{max} = 35°$, $\lambda_{max} = 35°$ |
| Gearbox characteristics | $x_R = 110$ mm, $a_R = 162$ mm, $b_R = 122$ mm |
| Max allowed torque | $M_{m,max} = 200$ Nm |
| Max allowed radial and axial forces | $Fr_{R,max} = 5000$ N, $Fa_{R,max} = 1000$ N |
| Allowed lever distance | $r_{min} = 100$ mm |
| Payload mass and center of inertia | $m = 250$ kg, $\boldsymbol{C_{in}} = \begin{bmatrix} 0 & 0 & 500 \end{bmatrix}^T$ mm |

Settings for RCMIGA are shown in Table 4. These values were chosen as most suitable for this problem after comparative analysis with different values.

**Table 4.** Settings for the Genetic Algorithm.

| Option | Value |
|---|---|
| Population size | 5000 |
| Elite count | 500 |
| Crossover fraction | 0.3 |

Optimization stopping criteria are shown in Table 5.

**Table 5.** Settings for stopping criteria of optimization.

| Criterion | Value |
|---|---|
| Max generations | 1000 |
| Max stall generations | 200 |
| Max time | 12 h |
| Max stall time | 1 h |
| Desired fitness function value | 0 |
| Fitness function value tolerance | $10^{-9}$ |

## 7. Results

The obtained values of the geometric parameters for the final optimization solution are given in Table 6. This is the obtained optimal solution for the values which are adopted and shown in Table 3 and for the chosen implementation of the fitness function (Equation (47)).

**Table 6.** Values of geometric parameters for the final solution.

| Parameter | Value |
|---|---|
| $a$ | 250 mm |
| $s$ | 1000 mm |
| $R_p$ | 490 mm |
| $R_b$ | 700 mm |
| $\gamma_p$ | 44° |
| $\gamma_b$ | 25° |
| $\varepsilon$ | 39° |
| $\tau_{ll}$ | 19° |
| $\tau_{jp}$ | −81° |
| $\mu_{ll}$ | −31° |
| $\mu_{jp}$ | 14° |

It should be noted that optimization settings and fitness function implementation can be easily adapted for other applications.

This paper's primary result is an algorithm capable of designing mechanisms with sufficient characteristics for flight simulation, taking into account all imposed constraints. Starting from values of geometric parameters that are generated based on a random uniform distribution with integer values within lower and upper bounds (later mapped to real values of geometric parameters), the algorithm has obtained a solution that can be manufactured and used for flight simulation.

Termination of optimization occurred because the max stall generation stopping criterion was reached. As shown in Figure 14, optimization stopped before reaching 600 generations, with the best penalty value of 52.842. Optimization found a solution that satisfies all constraints after five generations and reached a penalty value of 55 after the first 100 generations.

The kinematic model of optimized mechanism geometry is shown in Figure 15, and it is verified that this geometry of the mechanism has values of performance indicators the same as they are calculated within the optimization process. Figures 11 and 13 show the workspace for this particular set of geometric parameters.

Optimization began without any assumption or known solution. In the first generation, values of geometric parameters for each individual of the population are generated using a random uniform distribution. In the first five generations, none of the 5000 individuals in the population satisfied implemented nonlinear constraints. In the sixth generation, the first individuals satisfied all constraints and obtained the penalty value as a result of the evaluation of the fitness function. This search for individuals capable of satisfying all constraints is possible thanks to the implementation of crossover and mutation functions. Later these functions, together with elitism and selection, enabled convergence towards the final result, as shown in Figure 14.

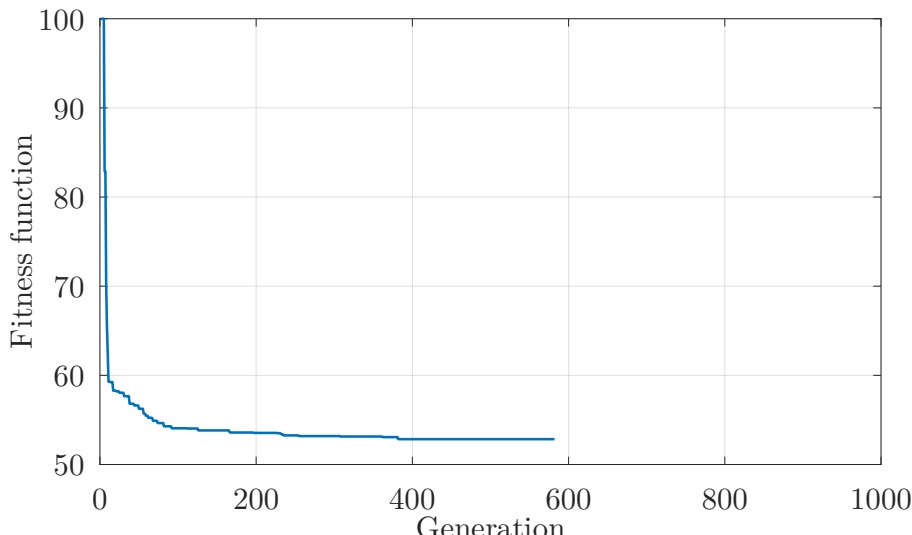

**Figure 14.** Change of penalty value (score) in reference to generation.

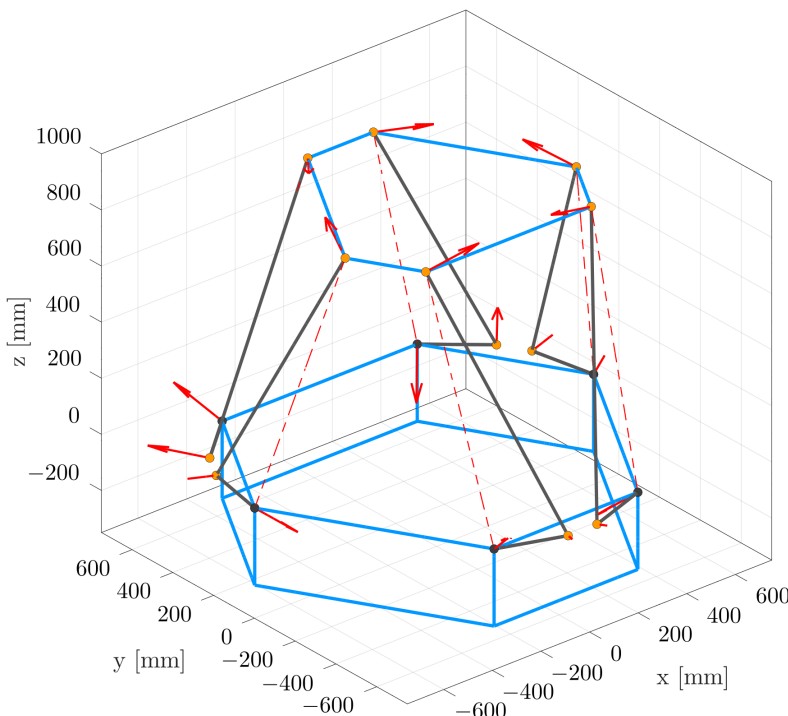

**Figure 15.** Kinematic model of optimized mechanism geometry.

Figure 16a shows the maximum required actuator torque in the $O_{xy}$ plane when all other external coordinates have zero value. Figure 16b shows the value of the dexterity indicator in the same plane.

Figure 17a,b shows maximal reachable values of pitch and roll angles in the $O_{xy}$ plane when all other external coordinates have zero value. As expected, the mechanism has the best performance near the home position, and its capabilities are reduced by changing the value of any external coordinate.

The following two tables (Tables 7 and 8) show maximal and minimal values of each external coordinate. These values are maximal and minimal values of the particular external coordinate that the platform can reach (a solution of inverse kinematics exists). During this calculation, only coordinate z was variable, and all other external coordinates had zero value. These results show the capabilities of the obtained mechanism.

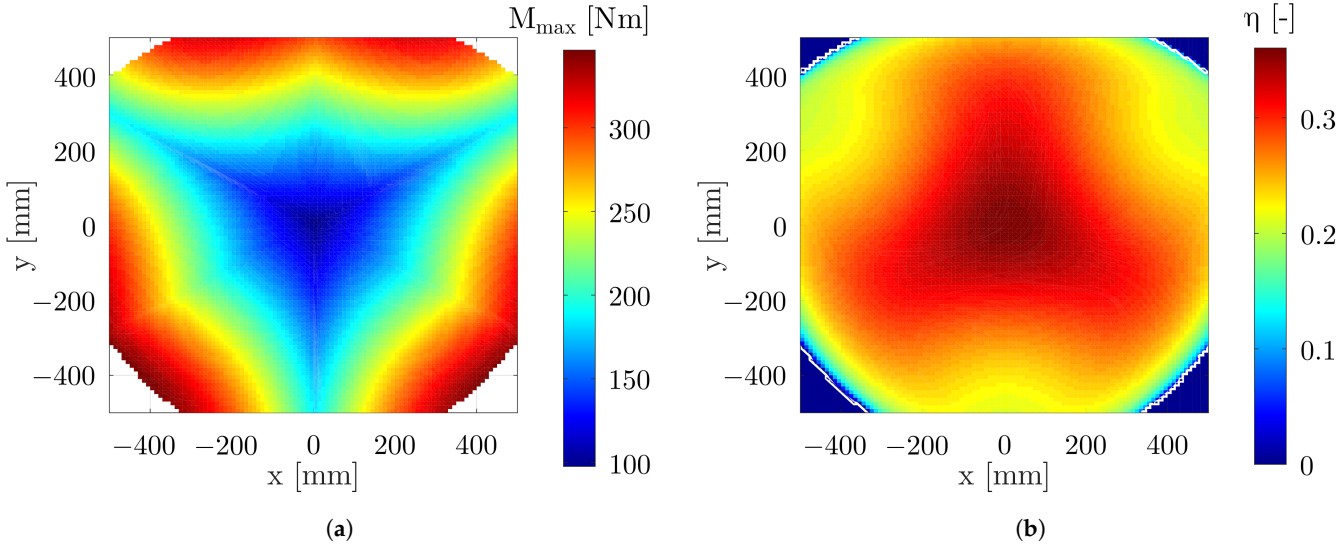

**Figure 16.** (**a**) Maximal required actuator torque and (**b**) dexterity of mechanism.

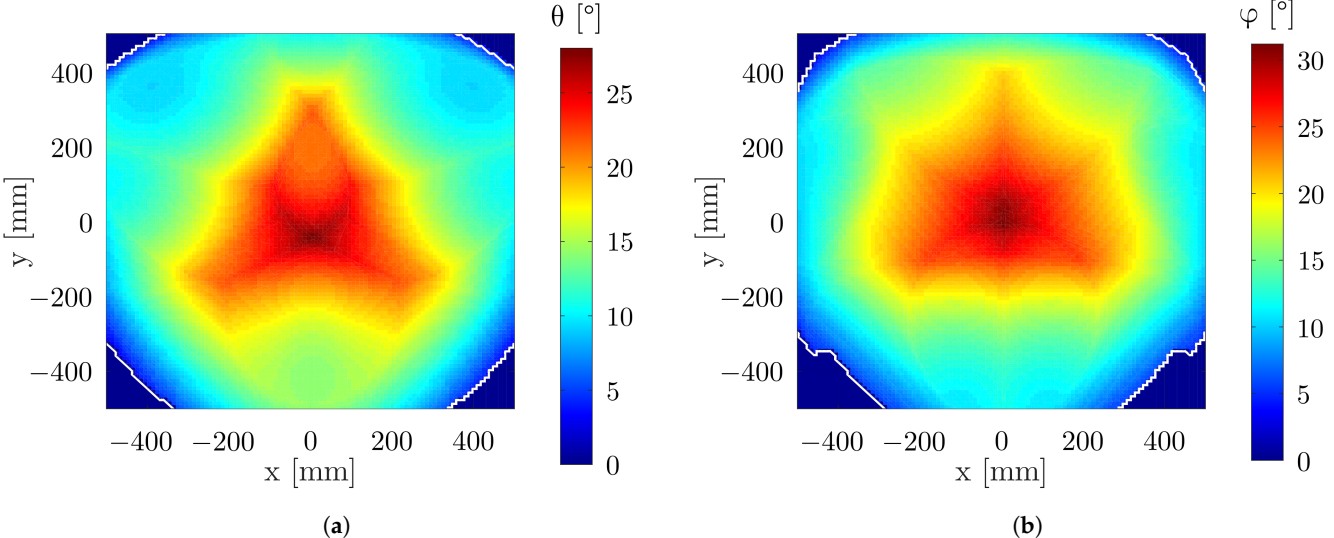

**Figure 17.** (**a**) Maximal pitch and (**b**) roll angles.

**Table 7.** Maximal values of external coordinates.

| External Coordinate | Value | Coordinate $z$ |
|:---:|:---:|:---:|
| $x$ | 640 mm | −38 mm |
| $y$ | 670 mm | −91 mm |
| $z$ | 290 mm | 290 mm |
| $\psi$ | 67.5° | 43 mm |
| $\theta$ | 31.5° | 33 mm |
| $\varphi$ | 38.5° | −6 mm |

**Table 8.** Minimal values of external coordinates.

| External Coordinate | Value | Coordinate $z$ |
|:---:|:---:|:---:|
| $x$ | −640 mm | −38 mm |
| $y$ | −750 mm | −89 mm |
| $z$ | −180 mm | −180 mm |
| $\psi$ | −67.5° | 43 mm |
| $\theta$ | −31.5° | 33 mm |
| $\varphi$ | −35° | 78 mm |

In order to experimentally verify the characteristics of the mechanism and everything that was analyzed in this paper, a first scaled-down prototype of the flight simulator mechanism was made (shown in Figure 18).

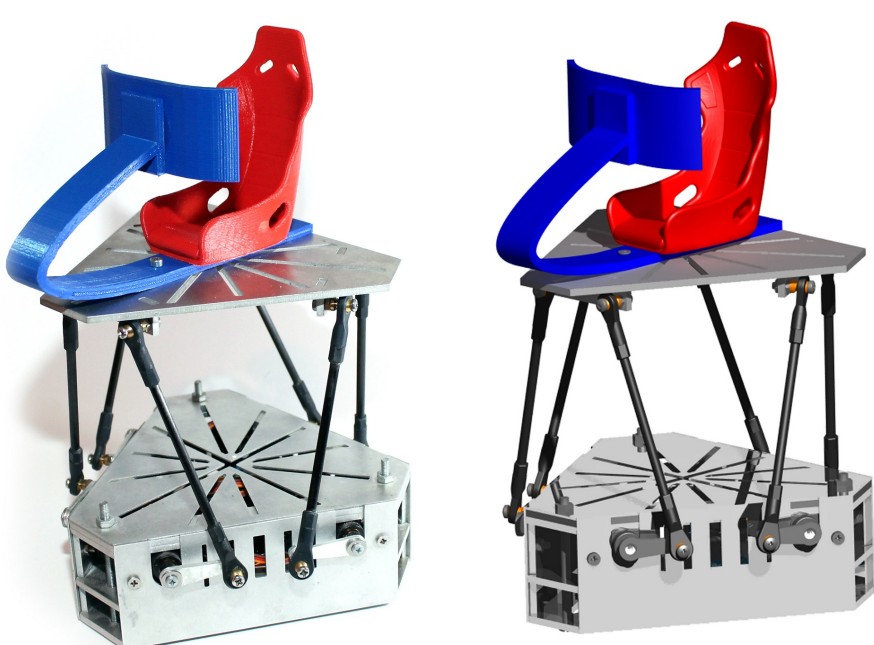

**Figure 18.** Scaled-down prototype of the mechanism (physical and 3D model).

Based on the presented solution of the inverse kinematic problem, all equations are implemented in the microcontroller of the control system. As part of the verification process, the platform moved in accordance with the desired movement, which was preprogrammed. All the assumed physical limitations of the mechanism also proved to be correctly implemented, and the calculated workspace corresponds to the one actually obtained. Using different sensors, it is possible to measure the positioning precision of the platform. It was first successfully tested whether actuators are able to achieve the desired angle based on the sent control value. Experiments with the assembled mechanism were based on sending control values to the mechanism and measuring the position and orientation of the platform in order to compare it with calculated values. This method is used to test each position and orientation within the workspace.

The motion monitoring system for the Stewart platform is presented in paper [62], and it can be used for verification. There are also vision-based control systems, as described in [63] that can also be used for monitoring and verification of motion.

When the mechanism is geometrically unambiguously defined, it is possible to move on to its materialization, in the sense of transforming a kinematic model into a three-dimensional model with all the accompanying information on the basis of which it can be manufactured. The beginning and the end of the materialization process are shown in Figure 19.

Besides physical experiments, for research and verification, it is very useful to do computer-based simulations. In paper [64], computer-aided design (CAD) software is used for visualization of the workspace, while papers [65,66] presented simulation environments for this type of mechanism.

After all the programs were tested on a scaled-down prototype of the flight mechanism simulator and after the optimized geometry was obtained, a functional full-size prototype of the flight simulator mechanism is developed and manufactured with sufficient precision to match obtained values of geometric parameters. This prototype of a low-cost flight simulator with electric rotary actuators and optimized geometry for flight simulation is then used for all experiments. The mechanism for different control values is shown in Figure 20.

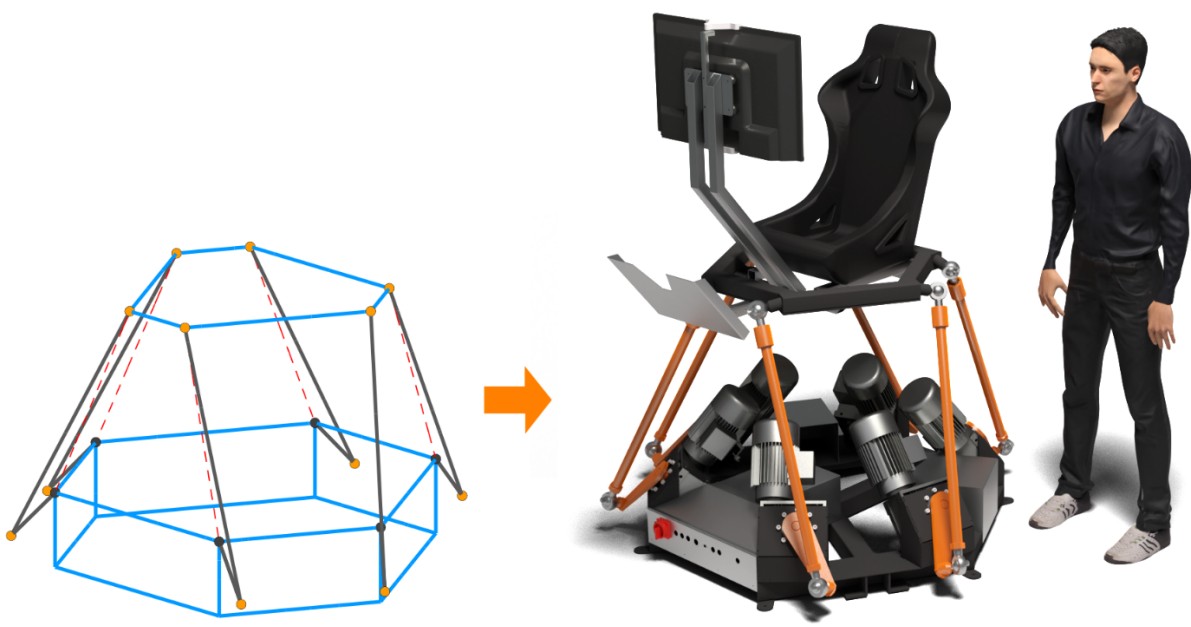

**Figure 19.** Materialization of the mechanism.

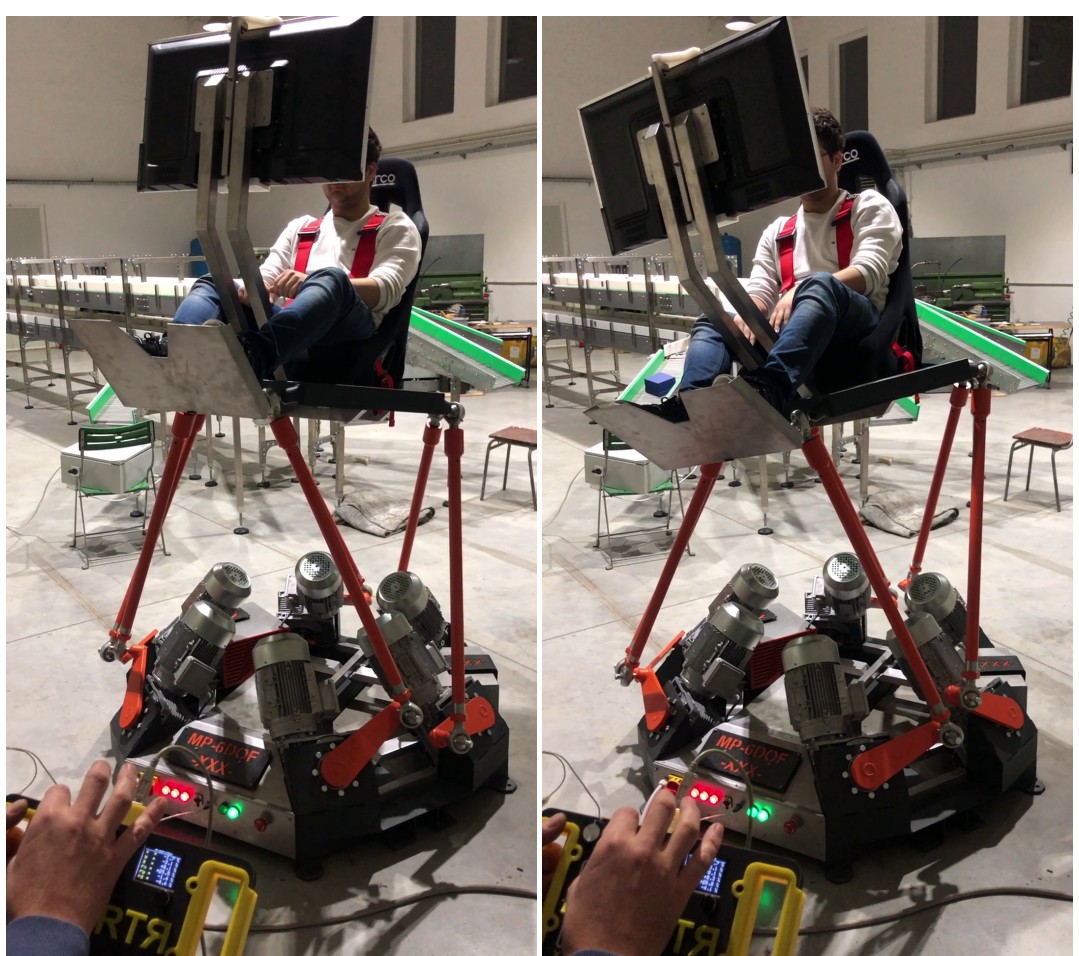

**Figure 20.** Experiment with a prototype of an optimized flight simulator mechanism.

This system can be used with flight simulation software that supplies the system with necessary motion data, or it can be used for a replay of captured data from the flight, but it can also be manually controlled. The system proved capable of flight simulation after

testing with preprogrammed trajectory characteristic for flight simulation in accordance with the most used airplane maneuvers.

### 8. Conclusions

In this paper, an algorithm for obtaining optimal geometry for the mechanism of the flight simulator based on the Stewart platform with rotary actuators is shown. There are algorithms that can be used in cases when the simulator cannot reach a particular position due to saturation of some degrees of freedom, to provide the feeling as if the real motion has been achieved (washout algorithms). Besides that, a good workspace and dynamic response are still a critical part of the design. Very important are the eleven geometric parameters which are proposed. They can fully define the geometry of the simulator with all constraints. Optimal values of geometric parameters were obtained using the real coded mixed integer genetic algorithm with all constraints and the appropriate fitness function that can be defined to suit many different design criteria and requirements. The considered optimization process is time efficient and can take advantage of parallel computing. No software library is required for determining geometry based on geometric parameters, for solving inverse and forward kinematics, or for the optimization algorithm. They can be implemented in any programming language using basic mathematical functions. Described algorithms are programmed, and they are fully automatic from the initial population to the final result.

It is shown that obtained geometry based on defined geometrical parameters can materialize, and two different prototypes were successfully developed. All obtained results and proposed algorithms are experimentally tested on the scaled-down model and on a full-scale prototype. A whole electrical system is developed for these prototypes, including angle sensing for lower levers and high-speed control of variable frequency drives. Developed programs for microcontrollers and desktop applications for sending configuration parameters and desired commands for the system are equally important.

Shown solutions for inverse and kinematics problems of the rotary Stewart platform in the first placed allowed optimization of the geometry, but equally important are imposed constraints and chosen performance indicators for flight simulation.

The optimization approach shown in this paper is not specific to this mechanism and can be easily adapted to other types of parallel mechanisms mentioned earlier. Further works should consider other optimization algorithms and compare their performances.

**Author Contributions:** Conceptualization, M.D.P., A.M.G. and D.M.P.; methodology, M.D.P. and A.M.G.; software, M.D.P. and N.G.R.; validation, A.M.G. and D.M.P.; formal analysis, M.D.P. and M.G.P.; investigation, M.D.P., M.G.P. and N.G.R.; resources, M.D.P., A.M.G. and D.M.P.; data curation, M.D.P., M.G.P. and N.G.R.; writing—original draft preparation, M.D.P.; writing—review and editing, M.D.P. and A.M.G.; visualization, M.D.P. and N.G.R.; supervision, A.M.G. and D.M.P.; project administration, M.D.P. All authors have read and agreed to the published version of the manuscript.

**Funding:** This research received no external funding.

**Institutional Review Board Statement:** Not applicable.

**Informed Consent Statement:** Not applicable.

**Data Availability Statement:** Not applicable.

**Conflicts of Interest:** The authors declare no conflict of interest.

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
