# Peer review of "Real Coded Mixed Integer Genetic Algorithm for Geometry Optimization of Flight Simulator Mechanism Based on Rotary Stewart Platform"

_applsci, doi:10.3390/app12147085_

Round 1

Reviewer 1 Report

              1.  The authors are recommended to revise the Abstract and consider the claim(s) of this study.

              2.  The authors are recommended to revise the Introduction by considering the importance and contributions of this study.  

              3.  The authors should clarify and discuss the novelty of the methodology more specifically in the Introduction.

              4.   This study suffers from a comprehensive reporting of the recent optimizers. The authors are recommended to boost the literature review by deep diving into recent metaheuristic algorithms like QANA: Quantum-based avian navigation optimizer algorithm and Starling murmuration optimizer: A novel bio-inspired algorithm for global and engineering optimization. The related studies must be addressed.

              5.   It is recommended to redraw and check Figure 10. Also, the conditional figures are needed the labeling “Yes or No”.

              6.  It is recommended to compare the performance of the real coded mixed integer genetic algorithm with some contender algorithms.

              7.  The statistical test and post hoc analysis are recommended.

              8.  The effectiveness of this study should evaluate using the effective formula in the DMDE: Diversity-maintained multi-trial vector differential evolution algorithm for non-decomposition large-scale global optimization.

              9.  The conclusion of this manuscript must be revised to highlight the findings of this study and the further works.

            10. Analysis of the results is missing in the paper. There is a big gap between the results and the conclusion. The analyses would be the core of the proposed model, where the authors prove their understanding of the reason behind the results. This paper requires a profound and comprehensive analysis of corresponding claims.

            11. It is suggested to show the name of the parameters in Figure 15.

            12. It is suggested to report all parameters used in the paper in the nomenclature table.

            13. It is recommended to formulate the problem and properties of this study. Why do the authors solve this problem using a metaheuristic algorithm? please clarify why brute force and exhaustive algorithms are not considered to solve this problem.

            14. The authors recommend providing a subsection of in-depth experimental analysis to investigate the impact analysis of claims. Also, the paper requires a very deep analysis from different perspectives.

            15. The convergence analysis is the main property of metaheuristic algorithms. Please visualised the convergence curves of algorithms.

            16. Please add the related work section for this study and insert Figures 1 and 2 in this section.

  17. It is suggested to add the Matlab code of this problem to the Appendix or GitHub.

Author Response

Dear editor,

We want to thank you for the detailed analysis of our paper and all the very useful recommendations and comments that you submitted to us in order to improve this paper. We have done a major revision according to your report.

Our answers are as follows.

1. In the Abstract, we first claim that we are applying a real coded mixed integer genetic algorithm (RCMIGA) for geometry optimization of the Stewart platform with rotary actuators (6-RUS) to design a mechanism with appropriate physical limitations of workspace and motion performances. As described in section 5, we successfully implemented a real coded mixed integer genetic algorithm for this problem without the use of any library, and we implemented all appropriate physical limitations of workspace and motion performances using constraints and upper and lower bounds of geometric parameters. The second claim is that the chosen algorithm proved that it can find the best global solution with all imposed constraints with a major accent on imposed constraints that can be implemented with this algorithm (subsection 6.1). It is shown that we obtained a solution. The third claim is that the obtained geometry can be manufactured because integer solutions can be mapped to available discrete values, which are also characteristic of the chosen algorithm. The last claim is that geometry is defined with a minimum number of parameters that fully define the mechanism with all constraints, which is described in section 2.

2. and 3. The importance, contributions of this study, and novelty of the methodology are now better explained in the Introduction.

4. We found recommended literature (three mentioned articles) significant for future works, and we added it as work on recent optimizers. Future works would include recent optimizers as in this paper is problem formulated, and it is shown that this particular optimization algorithm can be successfully applied to it.

5. Figure 10 is redrawn according to the recommendation.

6., 7. and 8. The goal of this paper is not to analyze the optimization algorithm but rather to apply it to the mechanical engineering problem of geometry definition of the Rotary Stewart platform for flight simulation, authors are Aerospace engineers, and our research is done in order to apply this type of mechanism for multiple applications. In future works, we will analyze other algorithms and compare their performances.

9. Findings and conclusions are better highlighted, and further works are added.

10. Analysis of the results is rewritten. Results are shown and analyzed in section 7. The result of this paper is an algorithm for the design of low-cost flight simulators with electric rotary actuators and optimized geometry for flight simulation. Satisfactory performance indices of the simulator mechanism are obtained, which means that problem is solved, and this mechanism can be used for flight simulation. Finally, the most important result is that, based on this paper, we successfully made a working flight simulator, as shown in the last figure. 

11. We do not understand which parameters you are referring to in Figure 15 (figure showing materialization of mechanism).

12. All geometric parameters are defined in section 2 and displayed in Table 1. It is hard to explain these parameters without additional figures, which is why the nomenclature table is not used and why we dedicated a whole section to it and explained everything in this section.

13. In the Introduction, it is explained that due to many different combinations and their interrelated influence on the simulator performance, it is challenging to find optimal values for these parameters. Metaheuristic algorithm is used in order to lower the number of combinations (computation of fitness function for this problem is very computationally expensive because of several million times of inverse and forward kinematics solving for each combination). Brute force and exhaustive algorithms are not used because of the computational power and time necessary for solving.

14. Added explanation about experimental analysis. As stated in the paper, experiments are used to check if the assumed physical limitations of the mechanism are implemented correctly and if the calculated workspace corresponds to the one actually obtained.

15. The change of fitness function in reference to generation is now shown in Figure 14.

16. Figures 1 and 2 are basically introductions to parallel mechanism and the rotary Stewart platform.

17. For this paper, the problem is not implemented in Matlab (the first conference paper used the Matlab toolbox for the genetic algorithms), and it is fully described in order for the reader to be able to obtain the same results in any programming language. 

Best regards,
authors

Reviewer 2 Report

the explanations about the geometry, especially figures 5, 6  are very unclear and hard to follow.  Figure 5 should have only angles / segments in the base plane, add on to it in figure 6 for a perpective 3d view the other relevant angles. 

It is not clear why you need integer optimization algorithms, what variables of those 11 in table 1 need to be integers ? Of course, if optimization says 230.1123 mm, you will round up to the reasonable precision, but why can a leg of length s of 1110.55 mm cannot be manufactured ?  

Author Response

Dear editor,

We want to thank you for your comments and questions. They showed us what we must improve and what we must explain better to make progress on our paper. We have done a major revision according to your report.

Figures 5 and 6 are both split into two figures in order to better explain parameters and also we changed text order in order to be easier to follow. Definition of actuator axis is moved after figure 7. We defined these geometric parameters this way in order to be able to optimize geometry with a minimal number of variable parameters and their definition is not very intuitive (primarily because of polar coordinates) and may be hard to follow but this is one of the most important contributions of this paper.

We completely changed section 2 to explain geometric parameters better.

It is not needed to use an integer optimization algorithm, but this is the most advanced type for this problem as you can precisely define the allowed precision of each geometric parameter without the need for rounding results. Mentioned length s of 1110.55 mm can be one of the allowed values (the allowed step can be defined as 0.01 mm if that is the precision of the manufacturing process). The optimization process uses integer values, but those values are then mapped to real values of parameters (which can be 1110.55 mm). With an integer optimization algorithm, search space is much smaller, which is important for computationally expensive problems such as this one. 

We added a better explanation for this in section 5.

Best regards,
authors

Reviewer 3 Report

Dear authors,
The article concerns the multi-criteria optimization issue with the use of genetic algorithms. The article is well-written, and the optimization problem is defined correctly. The list of references is sufficient. The introduction is correct.
I have just a few detailed questions:

1. What were the criteria for selecting the initial population? Were the values entirely random?
2. On what basis was the population size (5000) selected?
3. What crossover and mutation probability did you apply in your genetic algorithm implementation? There is a 0.3 value in table 4. Is this the probability of crossover (30%)? Usually, the probability of crossover is 20-40%, and the probability of mutation is about 4-6%.
4. How many times did you run your optimization algorithm?

Best regards,

Author Response

Dear editor,

We want to thank you for your comments and detailed questions. We improved the description of the method.
Our answers are as follows.

1. The first generation of the population is generated based on a random uniform distribution with integer values within lower and upper bounds (later mapped to allowed real values of geometric parameters).

2. It is selected after a few optimization runs based on convergence speed and time of calculations for each generation.

3. Yes, crossover fraction 0.3 means that 30% of the generation is used for crossover.

4. Values for optimization settings were chosen as most suitable for this problem after comparative analysis with different values, this analysis requested atleast a few dozen runs of the optimization algorithm.

Best regards,
authors

Round 2

Reviewer 1 Report

The authors have responded to most of my comments and I am confident that the revised manuscript can be accepted.

Author Response

Dear editor,

We want to thank you for your very useful comments and for taking the time to review our manuscript.

Best regards,
authors

Reviewer 2 Report

I think there is sufficient improvement in the presentation and the methods described, to publish this. It is still too complicated, in my opinion, and sometimes unclear. It is not an easy read, but it is interesting, nevertheless. It should  emphasize better the 120 degree symmetry, for example. Also, the implicit system that solve the inverse problems could be more clear.  Maybe it could be shortened somewhat, a reader could lose the patience.  

"necessary" is still spelled wrong in some place. 

Maybe some mention of the computer used to do the calculation, optimization, how much was of the process was  "manual" , how much was automated ? was there a software library used, maybe I missed the mention ?  

Author Response

Dear Reviewer,

We would like to thank you for taking the time to review our manuscript. We are grateful for your support and for your recommendation to publish this manuscript. We have done a minor revision according to your report.

Our answers are as follows.

- In order to define all Ai, Bi and Pi points, we need 6*3*3 = 54 components for the position in the Oxyz frame, and for the actuator's axis of rotation and axes of joints, we need to define 6*3 = 18 vectors. All of this is fully defined with just 11 variable parameters. That is the root of the complication. It may be hard to follow but on the other hand, it is very easy to implement it for optimization and computer calculations (which is the actual goal of this paper). It is necessary once to implement this algorithm for calculating position and axes vectors based on geometric parameters, and then it is as simple as changing 11 numbers. This explanation is now added to the manuscript.

- Added explanation about 120 degree symmetry.

- Added that equation 17 is the actual explicit solution to the inverse kinematic problem.

- Fixed wrong spelling of the word "necessary" in some places and a few other mistakes. 

- The computer used is not mentioned as it can be done on any digital computer. We did not present any time of computation or other performance indicators as they are irrelevant. The optimization algorithm is automatic without any manual stage. It is explained that even values in the initial population are obtained based on uniform random distribution. It is not mentioned, but we implemented the whole algorithm in several programming languages (GNU Octave, Matlab, JavaScript, C, C++), and no library is needed except for basic mathematical functions. This explanation is now added to the conclusions of the manuscript.

Best regards,
authors